# A framework for mutational signature analysis based on DNA shape parameters

Aleksandra Karolak[1,2¤], Jurica Levatić[1], Fran Supek[1,3]*

**1** Genome Data Science, Institute for Research in Biomedicine (IRB Barcelona), Barcelona Institute of Science and Technology, Barcelona, Spain, **2** Department of Population Sciences and Department of Computational and Quantitative Medicine, Division of Mathematical Oncology, Beckman Research Institute, City of Hope, Duarte, CA, United States of America, **3** Catalan Institution for Research and Advanced Studies (ICREA), Barcelona, Spain

¤ Current address: Division of Quantitative Science, Department of Machine Learning, Moffitt Cancer Center and Research Institute, Tampa, FL, United States of America
* fran.supek@irbbarcelona.org

**Data Availability Statement:** All relevant data are within the paper and its Supporting Information files.

**Funding:** This work has received funding from the European Union's Framework Programme for

## Abstract

The mutation risk of a DNA locus depends on its oligonucleotide context. In turn, mutability of oligonucleotides varies across individuals, due to exposure to mutagenic agents or due to variable efficiency and/or accuracy of DNA repair. Such variability is captured by mutational signatures, a mathematical construct obtained by a deconvolution of mutation frequency spectra across individuals. There is a need to enhance methods for inferring mutational signatures to make better use of sparse mutation data (e.g., resulting from exome sequencing of cancers), to facilitate insight into underlying biological mechanisms, and to provide more accurate mutation rate baselines for inferring positive and negative selection. We propose a conceptualization of mutational signatures that represents oligonucleotides *via* descriptors of DNA conformation: base pair, base pair step, and minor groove width parameters. We demonstrate how such DNA structural parameters can accurately predict mutation occurrence due to DNA repair failures or due to exposure to diverse mutagens such as radiation, chemical exposure, and the APOBEC cytosine deaminase enzymes. Furthermore, the mutation frequency of DNA oligomers classed by structural features can accurately capture systematic variability in mutagenesis of >1,000 tumors originating from diverse human tissues. A nonnegative matrix factorization was applied to mutation spectra stratified by DNA structural features, thereby extracting novel mutational signatures. Moreover, many of the known trinucleotide signatures were associated with an additional spectrum in the DNA structural descriptor space, which may aid interpretation and provide mechanistic insight. Overall, we suggest that the power of DNA sequence motif-based mutational signature analysis can be enhanced by drawing on DNA shape features.

## Introduction

Advances in analysis of mutation signatures are transforming genomics of cancer [1–5], human populations [6], and model organisms [7]. Tumor evolution is characterized by

Research and Innovation Horizon 2020 (2014-2020), under the Marie Skłodowska-Curie PROBIST grant agreement No. 754510 (to A.K. and J.L., PROBIST co-fund fellowship of the Barcelona Institute of Science and Technology) and by the European Research Council (ERC) under the European Union's Horizon 2020 research and innovation programme (grant agreement No. 757700, to F.S). F.S. is funded by the ICREA Research Professor program. A.K., J.L. and F.S. acknowledge support of the Severo Ochoa Centres of Excellence program of the Spanish Ministry of Economy and Competitiveness to the IRB Barcelona. Work in the laboratory of F.S. was supported by the ERDF/Spanish Ministry of Science, Innovation and Universities-Spanish Research State Agency/RegioMut project (grant agreement No. BFU2017-89833-P) and REPAIRSCAPE project (PID2020-118795GB-I00). The funders had no role in study design, data collection and analysis, decision to publish, or preparation of the manuscript.

**Competing interests:** The authors have declared that no competing interests exist.

distinctive somatic mutational processes resulting from mutagen exposures (environmental or endogenous) or defects in DNA repair mechanisms that result in genome instability [8–10]. Identification of these mutational processes can add to our knowledge of DNA damage and repair mechanisms that operate in human cells [11, 12]; it can contribute to understanding the etiology of various tumor types, with implications for predicting cancer risk [13, 14]; it can improve statistical methodologies for detecting cancer driver genes by refining baseline estimates of mutation rates [15, 16]; and finally, it has the potential to identify mutational biomarkers that can aid diagnostics [17, 18] and personalized treatment of tumors [19–21].

The genomic landscapes of individual cancers result from a combination of multiple overlapping mutational processes, making their deconvolution from genomic data a difficult challenge. There are many existing approaches to address this task [22], which apply a factorization technique to a frequency table of occurrences of mutations in various DNA contexts. The resulting mutational signatures can identify known examples of mutagenic mechanisms operative in certain cancer types [1, 2, 19]. However, many outstanding issues remain with methodologies for extracting mutational signatures. Firstly, many of the signatures were not matched with a mechanism or a clear biological covariate [1], which could represent novel biology but can often result from either incompletely/inaccurately resolved mixtures of mutational processes, or sequencing/alignment/mutation calling artefacts. Additionally, the existing mutational signature extractions do not appear very robust: various statistical approaches to infer mutational signatures do not necessarily extract consistent sets of mutational signatures, and moreover even with the same method, minor perturbations to the input data (e.g. same biological process across different tissues) can result in different extracted signatures. One reason for the lack of robustness is that the somatic mutation frequency data tend to be sparse; noise due to low mutation counts can overwhelm biological signal. This is aggravated when changing the tabulation of oligonucleotides from the commonly used trinucleotide (3 nt) DNA sequence representation, to longer, more informative representations–pentanucleotides (5 nt) or heptanucleotides (7 nt)–where the combinatorially increasing number of possible oligonucleotides aggravates sparseness. In addition to these statistical considerations, there are difficulties with interpreting the signatures: DNA sequence is usually not in obvious ways related to the biochemical aspects of the DNA damage and repair processes, and so the sequence-based mutational signatures do not facilitate insight into underlying mechanisms.

To address the challenges above, there may be benefits to enhancing the DNA oligomer representation for mutation signature analysis. Firstly, robustness (towards noise and systematic biases) of the methodologies for signature inference may be improved by reducing data sparseness. Secondly, the ability to interpret the signatures and link them to biological mechanisms may benefit. Thirdly, new representations can help identify additional mutational signatures that are not 'visible' to the standard trinucleotide/pentanucleotide approach. Here, we propose a framework to integrate information about DNA structure [23, 24] of the DNA oligomers to predict their mutability and to infer mutation signatures. We were inspired by the known examples of DNA structural features susceptible to certain mutagens, such as DNA hairpin structures vulnerable to the APOBEC3A cytosine deaminase [25], various other types of DNA repeats with tendency to form non-B-DNA conformations [26], high curvature of longer DNA segments that associates with mutation rates [27], or DNA structure changes upon AP-1 transcription factor binding that sensitizes to UV damage and consequently mutation [28, 29]. Our framework generalizes over many of these examples, employing a diverse set of ds DNA shape features to describe neighborhoods of mutated loci in human cancer. Our implementation utilizes precalculated base-pair, base-pair step, and minor groove shape parameters of DNA oligomers [30, 31]. Such structural parameters are considered to be an accurate description of DNA conformation, summarizing atomic coordinates of nucleotides

in a compact representation [32–34]. DNA susceptibility to mutagenic agents or recognition by DNA repair enzymes might be enhanced or disrupted by genetic differences in the regions flanking the mutation site [35, 36]. Because DNA predictably acquires a sequence-dependent local conformation, this provides a rationale for implementing sequence-derived DNA shape parameters into a framework to predict and classify mutagenic processes.

## Materials and methods

### Obtaining somatic mutations from cancer genomic data

We extracted somatic single-nucleotide variants (SNVs; henceforth: mutations) from the whole-genome sequences (WGS) of tumors from 30 cancer types; we did not consider indels or structural variation. The called mutations from cancer WGS were collected from: (i) The Cancer Genome Atlas (TCGA, https://www.cancer.gov/tcga), studies: BLCA, BRCA, CESC, COAD, DLBC, GBM, HNSC, KICH, KIRC, KIRP, LGG, LIHC, LUAD, LUSC, OV, PRAD, READ, SARC, SKCM, STAD, THCA, UCEC; (ii) International Cancer Genome Consortium data portal (ICGC, https://icgc.org), including somatic mutations from studies CLLE-ES, ESA-D-UK, LIRI-JP, MALY-DE, MELA-AU, PACA-IT, RECA-EU; and (iii) samples downloaded from the websites of individual WGS study of MDBA [37]. The final list included over 1,600 tumor samples, for which the chromosome number and coordinate of each somatic mutation were extracted using R 3.6 [38]. Next, DNA motifs up to +/- three nucleotides flanking each mutation site were retrieved using human GRCh37/hg19 as reference genome. This resulted in tri, penta, and heptanucleotide sequences ("3nt", "5nt", "7nt", respectively) with the mutation placed in the central position of the oligonucleotide. For the Poisson regression analyses (see below), all three types of DNA sequence motifs were further processed to count their mutation frequency in each tumor sample, and, as a baseline, their overall occurrence in the human genome. For the Principal Component (PC) analysis, each set of mutations (sorted by tri, penta or heptanucleotide) in a given tumor sample was further separated into six mutation outcomes: C>A, C>G, C>T, T>A, T>C and T>G (these are equivalent to, and were considered together with G>T, G>C, G>A, A>T, A>G and A>C, respectively, due to DNA strand symmetry).

### DNA representation using structural features

Within pentanucleotide DNA motifs, we extracted various structural parameters thereof. This produced: one minor groove width parameter defined for the central nucleotide (*mgw*0); six base pair (bp) parameters: propeller (*prop*0), opening (*open*0), buckle (*buck*0), stretch (*stre*0), stagger (*stag*0), shear (*shea*0); and additionally six base-pair step parameters, each preceding and following the central nucleotide (thus times two), resulting in total of twelve base-pair step parameters: twist (*tw*-1, *tw*+1), roll (*ro*-1, *ro*+1), tilt (*ti*-1, *ti*+1), slide (*sl*-1, *sl*+1), rise (*ri*-1, *ri*+1), and shift (*sh*-1, *sh*+1) [30]. DNA shape parameters extraction was performed using DNA-shape R routine [39, 40], which uses the Curves+ algorithm [41]. The total number of parameters increased from nineteen (1+6+12) for the pentanucleotide-based DNA structure representation ("5nt-str") to forty five for the heptanucleotide-based DNA structure representation (3+18+24; "7nt-str"). Although all shape parameters including *mgw*, 6 base pair parameters, and 6 base pair step parameters could in principle generate predictive features for mutation risk, four of these (*mgw*, *twist*, *roll*, and *propeller*) were previously considered more relevant for evaluating protein-DNA interactions [42–45] and thus by analogy we focus on these features also for mutation risk prediction. Additionally, using a more focused set of DNA structure features allowed us to match the number of features on the sequence-context side, and to reduce the number of free parameters in the signature analyses. An illustration of the

relationship between nucleotide position and the DNA structural parameters is shown for minor groove width, base-pair parameter propeller, and the base-pair step parameter roll in Fig 1. After extraction of the most informative parameters: minor groove width, propeller, twist, and roll, values of each parameter were normalized to range from 0 to 1, and discretized into three equally distributed bins: high ("H"), medium ("M"), and low ("L"), based on the known, nearly symmetrical distributions of the shape parameters around their equilibrium values [31].

## Modelling mutation counts in oligonucleotides by Poisson regression

The DNA sequence features on the one hand, and DNA shape parameters on the other hand, were applied to the task of predicting the mutation propensity of each oligonucleotide, for six individual hypermutated tumor samples (described below in the Results section), using count modelling and in particular the Poisson regression analysis as implemented in the R environment via the *glm* function [46]. Poisson regression is commonly used for modeling count data. The model is given by:

$$log\ \lambda_i = \beta_0 + \beta_1 x_{i1} + \ldots + \beta_p x_{ip}, \tag{1}$$

where $\lambda_i$ is a countable response variable, modelled as the linear combination of the covariates corresponding to the $i^{th}$ observation of the predictor variable $x$. Similarly to linear and logistic regression, the covariates are fixed, and the regression coefficients $\beta = (\beta_0, \ldots, \beta_p)$ are the model parameters to be estimated. The countable response variable $\lambda$ can be substituted with the ratio $\lambda/D$ of the count of the events (here mutations) and total number of opportunities $D$ that the event had to occur. This leads to an extra term–an offset *log D*–added on the right-hand site of the equation and an adjustment of the regression coefficients.

For DNA sequence features, particular each trinucleotide or pentanucleotide was a data point in the regression and was described by (i) the DNA sequence motif features (henceforth referred to as "3nt-seq" or "5nt-seq", respectively; these are simply variables indicating occurrence of A, G, C or T at each position; the motifs are considered DNA strand-symmetrically i.e. the reverse-complementary tri/pentanucleotides are collapsed together, thus the central nucleotide can have only 2 [instead of 4] values which were here chosen to be C and T to keep with the convention in mutation signature analysis), (ii) the mutation count of the tri/pentanucleotide in that tumor sample, and (iii) its occurrence within the human reference genome GRCh37/hg19 (the latter was introduced as an offset value into the regression, thus adjusting for the differential occurrence of oligonucleotides in the genome sequence). For DNA structural features, the sets of all structural parameters extracted for pentanucleotide and heptanucleotides (henceforth, "5nt-str" and "7nt-str"), each divided into L, M, and H bins, was used in

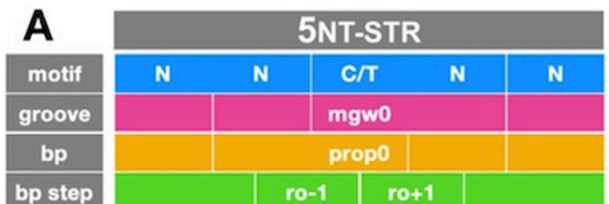
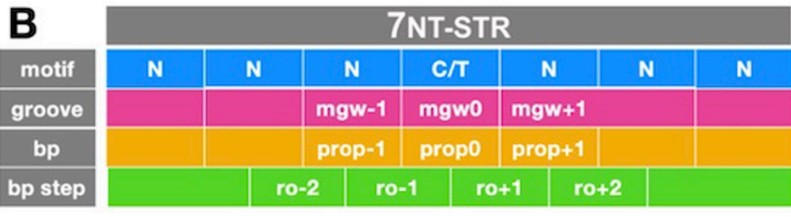

**Fig 1. Mapping of the structural DNA parameters on DNA sequence.** N can be any of A (adenine), C (cytosine), G (guanine), or T (thymine). Structural features considered: minor groove width (*mgw*), base-pair parameters (bp, shown example for *propeller*), and base-pair step parameters (bp step, shown example for *roll*) and their position within the (A) 5nt-long and (B) 7nt-long DNA motifs. Considering the DNA pentamer sequence is an approach to derive DNA structural features at a locus, while accounting for the nearest and next-nearest neighbors of the central nucleotide. The structural features: *mgw*, the one bp, and the two bp step parameters, at a given central base pair are the function of its pentamer environment.

place of the 3nt-seq or 5nt-seq in the DNA sequence analysis. The mutation type is not explicitly considered in the Poisson regression as implemented, but instead all mutations occurring at a C:G pair are considered jointly (C>A, C>T and C>G), and all mutations considered at an A:T pair are considered jointly (T>A, T>C and T>G). The assumption underlying this is that in a tumor predominantly affected by a single mutagenic process, different mutation types are likely to originate from the same or a shared mechanism. This provides a rationale for treating various mutation types as a single set, thus increasing statistical power to find associations. Of note, the more commonly observed mutation types in a particular tumor sample will have a higher weight towards determining coefficients in the regression run on that tumor sample. The McFadden pseudo-$R^2$ (henceforth, $pR^2$) statistic was used to evaluate model performance (fit to data), for each set of features and for each of the six tumors. All Poisson regression analyses and visualization were performed with R 3.6 [38].

## Estimating systematic variation in mutational patterns by factorization techniques

A principal component (PC) analysis was performed jointly for blocks of features describing the DNA sequence, and features describing DNA structure. On the DNA sequence side, 96 mutation contexts were considered within each tumor sample. In order to keep the number of structural features the same, thus making the estimations of relative contributions of sequence and shape parts toward each principal component balanced, we prepared 96 mutation contexts on the DNA shape side. This was achieved by focusing only on the L/M/H binned (see above) features of the six DNA structure parameters with the highest potential for interpretation: *mgw*0, *prop*0, *ro-1*, *ro+1*, *tw-1*, and *tw+1*. Six parameters could sample a conformational space assigned to one of the three bins: L, M, and H, generating 18 features, which were examined separately for six mutation types (C>A, C>G, C>T, T>A, T>C, T>G; considered DNA strand symmetrically) and thereby a total of one hundred eight contexts were obtained. After removing energetically forbidden (not sampled), conformational parameters: *prop*0*(L)* and *ro-1(H)* on the C sites, and *prop*0*(H)* and *tw+1(L)* on the T sites, the number 108 was reduced by 12 and resulted in 96 mutation contexts on the DNA structural side. Next, mutation counts in the structural block of each tumor sample were adjusted to match the sequence contribution to mutation burden in the corresponding sample, i.e., sampled to sum to the same values for the DNA sequence feature block and the DNA structure feature block. Additionally, the DNA structural features block was adjusted to have equal counts to the DNA sequence feature block, using a sampling function with the *UPmultinomial* function in R package *sampling* [38]. In this way, the noise due to low mutation counts will be equal in the two feature blocks and will not bias the PC analysis toward one block of features.

Our application of non-negative matrix factorization (NMF) to extract mutational signatures in parallel from trinucleotide mutational spectra and from DNA structural features (same set as for the PC analysis) was performed as follows. We extracted cancer cell line mutational signatures from 96 component trinucleotide mutation spectra and 96 DNA structural features of WGS samples. To extract mutational signatures we used custom R [21] implementation of the non-negative matrix factorization (NMF) based methodology, broadly as described by Alexandrov et al. [1] From the matrix containing mutation spectra and DNA structural features of samples, we generated 300 bootstrap samples. One bootstrap sample is obtained with *sampling* function of the *UPmultinomial* R package [38] applied to each sample's spectrum. Next, we used the NMF algorithm to each of the bootstrap samples (*nmf* function of the *nmfgpu4R* R package) to get different NMF runs; we used the *Multiplicative update rules* algorithm [47] with 10000 as the maximal number of iterations). For each bootstrap sample,

we varied the number of signatures. We used the 'hierarchical extraction' procedure proposed by Alexandrov et al. [1] where NMF is iteratively repeated while removing the well-reconstructed samples (cosine similarity above 0.97) from a previous iteration to discover new signatures. We allowed a maximum of 3 iterations.

From all the candidate mutational signatures obtained, we first searched for the signatures that closely resembled the ones that were previously found in human cancers [1] (referred to as PCAWG signatures). We compared the individual signatures obtained by all the different runs of NMF (given the different bootstrap samples and the different number of signatures) to the PCAWG signatures. For this comparison we calculated the cosine similarity on the 96 trinucleotide spectra (the 96 DNA structural features were not used). For each PCAWG signature, we searched for the closest matching NMF run. As a final set of mutational signatures, for each PCAWG signature we kept the closest matching NMF run if its cosine similarity to the best matching PCAWG signature exceeded 0.85. We use the NMF scores as the signature exposures across different tumor samples.

This procedure yielded 54 mutational signatures. The obtained signatures are named according to the PCAWG signatures they resemble, e.g., the signature name SBS15/6L denotes this signature was the closest match to PCAWG SBS15 (i.e., SBS15 is the primary signature); 6L denotes that the signature also resembled PCAWG signatures SBS6 (cosine similarity > 0.85). The "SBS" stands for "single base substitution"; here we do not consider indels or structural variation signatures. The suffix "L" (for "like") denotes $0.85 \leq$ cosine similarity $< 0.95$ (a somewhat less-close match), while the absence of the suffix "L" means cosine similarity $\geq 0.95$. Names of signatures other than the primary signature (if present) are ordered by decreasing cosine similarity.

We next searched for signatures specific to our dataset, i.e., signatures that commonly appear in our data but do not resemble any of the known tumor signatures. To this end, we employed k-means clustering (*clara* function in *clusters* R package, with Euclidean distance, standardization, "pamLike" options and the number of samples to be drawn from the dataset set to 10%). Each batch of the NMF solutions (from the signature extraction method selected as final by the evaluation) obtained as described before (one batch consists of 300 x $n$ solutions, where $n$ is the number of signatures; varies from 2 to 40) was clustered into $k$ clusters with k-means clustering varying k from 2 to 40. We chose the clustering result (i.e., a set of signatures) where the agreement with PCAWG signatures was maximized in terms of the number of clusters medoids that resemble PCAWG signatures (at cosine similarity >0.85). From such a set of signatures we selected the ones dissimilar from any of PCAWG signatures (cosine similarity <0.8), yielding in total 6 structural signatures denoted as SBS-SS (SBS structural signature).

## Results

Based on the known individual examples of DNA structural features associated with mutation occurrence via specific mechanisms [25–28], we hypothesized that a broad set of diverse DNA structural features would be able to predict mutation rates due to many different mutagenic exposures. To test this hypothesis, we predicted mutational frequency of DNA segments using count models (Poisson regression). In brief, in this approach a certain DNA oligonucleotide can be defined either by the DNA sequence features (occurrence of A, G, C or T in each position of the flanks of a locus), or by the DNA structural features (listed in Methods), while adjusting for the total number of occurrences of that oligonucleotide in the human genome sequence. We selected six hypermutated tumor samples as representatives of important mutagenic processes: (i) microsatellite instability (MSI) caused by defective DNA mismatch repair (MMR) in a representative colorectal tumor sample; (ii) activity of the proofreading-deficient

DNA polymerase epsilon (POLE) in another colorectal cancer sample bearing a S297F hotspot mutation; (iii) a bladder tumor sample bearing the mutational signature of the APOBEC cytosine deaminase [23]; (iv) a lung adenocarcinoma sample highly enriched with the tobacco smoking mutational signature; (v) ultraviolet (UV) light-induced mutagenesis in a melanoma sample; and (vi) the hypermutation induced by therapy by the DNA methylating drug temozolomide (TMZ) in a glioblastoma sample. Such hypermutated tumors derive most of their mutations from a single mechanism. This allowed to examine the mutagenic processes individually and to identify the sequence features and/or structural features of highly mutable DNA oligomers. To check consistency of results across individuals, we considered additional samples affected by the same mutational processes (S1 Table and S1 Fig; 4 additional tumors per process, except for TMZ exposure, where there were 2 additional tumor samples available). As a negative control, we further considered tumor samples originating from matched tissues, but not affected by the single-process hypermutation (S1 Table and S1 Fig).

## Sequence and structural features of oligonucleotides that hypermutate upon DNA repair failures

The Poisson regression coefficients corresponding to DNA sequence features–derived from trinucleotide (Figs 2A and S1, "3nt-seq") and pentanucleotide (Fig 2B, "5nt-seq") neighborhoods–quantify the impact of flanking DNA sites on mutational risk of the central site. For example, in the MSI tumor (Fig 2A), the central cytosine, C, (or, equivalently, the guanine, G, it pairs with) is more mutable than the thymine, T (equivalently, adenine, A). Furthermore, a guanine in the immediate 3' flanking position ("+1" in Fig 2A, MSI tumor) is further associated with a higher mutation rate of the central nucleotide in this tumor, consistent with very

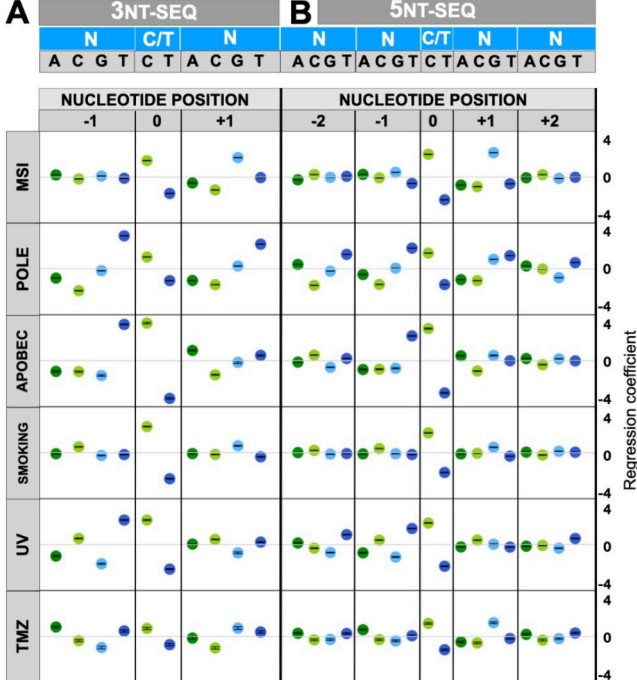

**Fig 2. DNA sequence as predictor of mutational burden in tumor samples by mutagens: MSI, POLE, APOBEC, smoking, UV, TMZ.** N can be any of the four DNA bases: A, C, G, or T (color coded respectively: dark green, light green, light blue, or dark blue). Poisson regression coefficients for (A) 3nt-seq, and (B) 5nt-seq. Error bars superimposed on each symbol show 95% C.I.

high mutation rates of the (commonly methylated) CpG dinucleotide in MMR-deficient cancers [48], and consistent with the mutational signature SBS6 previously detected for many colorectal cancers with MSI [1]. The extended pentanucleotide neighborhood appears to have more subtle associations with the MSI mutagenesis, at least in this particular MSI tumor; nonetheless some enrichment with cytosine at -2 and +2 positions was observed (Fig 2B).

Next, we turned to examine the DNA structural features that associate with mutation occurrence in the MSI tumor sample (top row, Fig 3A, "5nt-str"). We examined Poisson regression coefficients for the high "(H)", medium "(M)" and low "(L)" bins of each structural feature, as derived from the pentanucleotide neighborhoods (see Methods). Among other features, we observed a slight positive association with the *mgw0(H)* bin (pentanucleotides with high minor groove width at the central position), as well as with the *tw+1(L)* bin (a low value of twist parameter for the +1 position; see Fig 1 for schematic). A further analysis using structural features derived from an extended, heptanucleotide neighborhood ("7nt-str"; top row, Fig 3B) confirm the above and further suggest a narrowing of the minor groove at the position +1. To quantify the overall utility of DNA structural features for predicting mutation rates, we examined the overall fit of the model *via* the McFadden pseudo-$R^2$ statistic (p$R^2$; see Methods) (Tables 1 and S1). The DNA structural features exhibited a higher predictive ability (p$R^2$ = 0.60 for the *7nt-str*) compared to the composition DNA sequence features (highest p$R^2$ = 0.56 for the *5nt-seq*). We note that our shape features were computed from the DNA sequences by employing data resulting from previous simulations of DNA structures of various oligonucleotides (see Methods). This increase in model fit by using the DNA structural features indicates that they capture those statistical interactions between sites in a DNA motif that are relevant

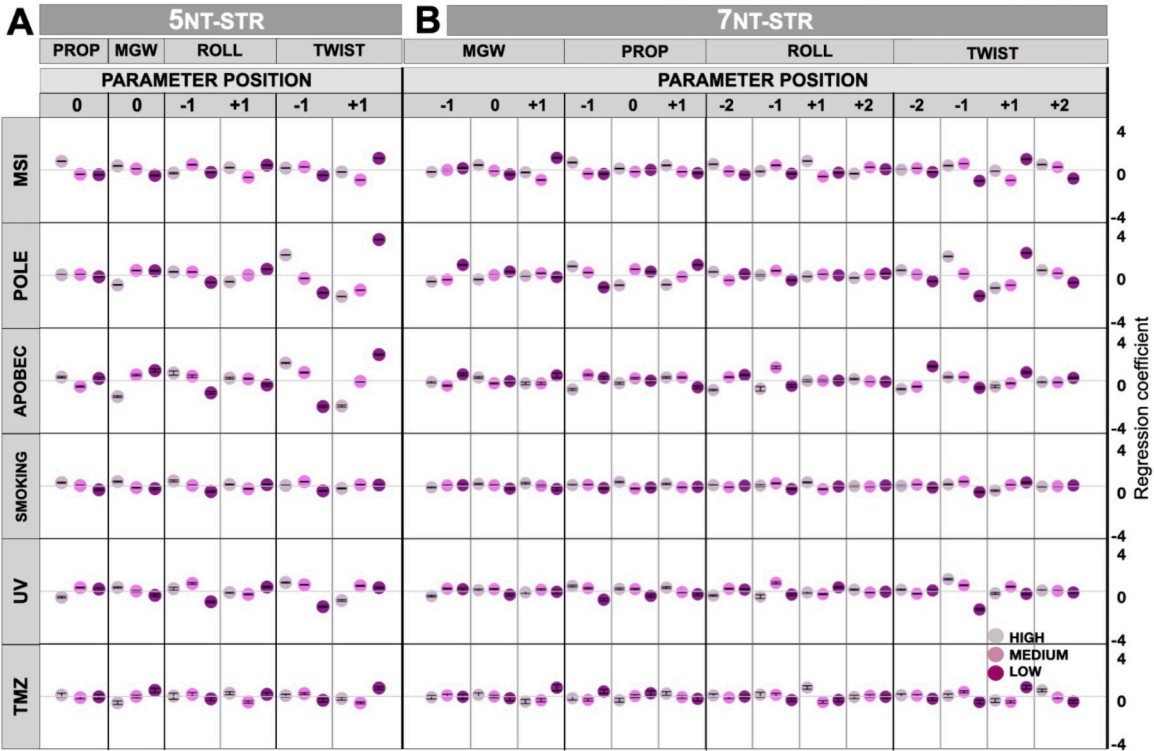

**Fig 3. Poisson regression coefficients describing mutation rates resulting from six mutagenic processes (MSI, POLE, APOBEC, SMOKING, UV and TMZ).** These were evaluated from (A) 5nt-str, and (B) 7nt-str representations. Parameters were normalized and divided into 3 equally spaced bins: high, medium, and low. Error bars are 95% C.I.

**Table 1. Performance of four models: 3nt-seq, 5nt-seq, 5nt-str, and 7nt-str.** Representative of distinct mutational processes, each row is a tumor sample with one strong mutational exposure.

| Predominant signature | Cancer type | Hypermutation type | McFadden R$^2$ | | | |
|:---:|:---:|:---:|:---:|:---:|:---:|:---:|
| | | | 3nt-seq | 5nt-seq | 5nt-str | 7nt-str |
| SBS6 | COAD | MSI | 0.57 | 0.56 | 0.55 | 0.60 |
| SBS10 | COAD | POLE | 0.86 | 0.81 | 0.76 | 0.81 |
| NA | BLCA | APOBEC | 0.94 | 0.92 | 0.77 | 0.85 |
| SBS4 | LUAD | SMOKING | 0.98 | 0.83 | 0.85 | 0.50 |
| SBS7 | SKCM | UV | 0.89 | 0.82 | 0.66 | 0.74 |
| NA | GBM | TMZ | 0.78 | 0.58 | 0.52 | 0.28 |

for predicting mutability of the oligomer. In other words, this representation of DNA shape preserves the important information for describing mutation rates while ignoring the less important information, suggesting that DNA structural descriptors may be a useful representation for mutational signature analysis. In addition, structural features highlighted by the regression (see above) may have potential for interpretation of mutagenic mechanisms in MMR-deficient, MSI tumor samples.

Next, we turned to examine mutations resulting from another sort of DNA repair failure common in tumors–deficient proofreading activity of the replicative DNA polymerase epsilon due to a mutation in the POLE gene [49]. In the POLE-signature enriched colorectal tumor sample (POLE row, Fig 2), the landscapes of regression coefficients from DNA sequence features contain many noticeable signals located further from the central nucleotide, consistent with previous work that suggested that up to nonanucleotide (9-mer) sized DNA motifs are informative for POLE mutagenesis [15]. We detected enrichments of thymine and to some degree adenine at -2, -1, +1 and at +2 positions next to the mutated central cytosine (Fig 2). The observed pattern resembles the mutational signatures SBS10a and SBS10b detected in colon and uterus cancers [1], further supporting the use of our methodology based on count models (here, Poisson regression) to model mutation risk. More interestingly for the matter at hand, the DNA structural analyses of the mutated loci in the POLE tumor (Fig 3) show strong signal in the twist at -1 and +1 positions, where DNA is overtwisted and under-twisted, respectively. This suggests there is a local deformation at the POLE hypermutable sites, tapering off towards a more regular shape further away from the center. As with the MSI tumor, also in the POLE tumor the structural DNA features yielded a similarly well-fitting model (pR$^2$ = 0.81 for *7nt-str*, Table 1) as the sequence DNA features did (pR$^2$ = 0.81 for *5nt-seq*). This further supported that the structural features we examined are appropriate to describe the propensity of DNA sites to mutate in DNA repair-deficient tumors. The overall predictive accuracy was higher for the POLE tumor than for MSI tumor (Tables 1 and S1). This suggests that mutational risk can be predicted from DNA shape to a variable degree across different mutational processes, similarly so as when predicting mutational risk from DNA sequence.

## Structural features associated with mutagenesis due to an endogenous DNA damaging activity of the APOBEC3A enzyme

In addition to examples of DNA repair deficiencies that result in hypermutation (MSI, POLE), we next turned to examining DNA structural features of sites mutated by DNA damaging agents. This included examples of both endogenous (APOBEC cytosine deaminases) or exogenous agents: radiation (UV) and chemicals (tobacco smoke, and the DNA methylating drug TMZ). Regression on trinucleotide contexts in the APOBEC-enriched bladder cancer sample

shows a very strong association of thymine at -1 position with mutation risk (Fig 2A, APOBEC row), as expected from known mutational signatures SBS2 and SBS13 [2]. The pentanucleotide regression (Fig 2B) finds enrichment of pyrimidines at position -2, supporting the role of the APOBEC3A enzyme [50] rather than the APOBEC3B paralog in mutagenizing this particular tumor. This is consistent with high propensity towards APOBEC3A mutagenesis in bladder cancer [51]. With respect to the DNA shape analysis, we note that APOBEC binding and deamination occurs on single stranded DNA, where DNA shape parameters have altered inter-pretation or are not well defined. However, the steps following the APOBEC-mediated DNA damage (i.e., cytosine to uracil conversion), involving repair by the base excision repair path-way (e.g., the UNG protein), can involve double-stranded DNA, making the structural features we examined also pertinent to APOBEC mutagenesis. The local tendency towards over-twist-ing at -1 and then under-twisting at +1 position is evident from the DNA structure coefficients (Fig 3A). Structural features in a broader DNA context (Fig 3B, *7nt-str*) additionally show that position -2 exhibits under-twisting at APOBEC3A mutated sites. Considered together with over- and under-twisting at positions -1 and +1, respectively, this suggests that DNA motifs experiencing APOBEC mutagenesis may be more prone to have flipped out bases. There are no notable changes in the roll parameter, suggesting it is not DNA bending that affects APO-BEC mutation frequency but rather the tendency toward exposing a single (here, central) nucleobase. Further supporting that a broader DNA oligomer context is predictive for APO-BEC mutagenesis, the model fit of the *7nt-str* descriptors is higher than for *5nt-str* descriptors (Table 1). We note that the simple trinucleotide representation of DNA sequence is highly pre-dictive (Table 1), probably reflecting the strict requirement for 3' T in APOBEC mutagenesis. The C>T APOBEC mutation type presented broadly similar associations with sequence and structural features to those of the C>G mutation type (S2 Fig).

## DNA structural features confer risk of mutation resulting from exogenous DNA damaging agents

Turning towards exogenous mutagenic agents, we examined a tobacco smoking-enriched lung adenocarcinoma sample. Such tumors are predominantly associated with the signature SBS4, consisting of C>A mutations in various trinucleotide contexts [1, 2]. In accordance with SBS4, the trinucleotide and pentanucleotide DNA sequence coefficients do not indicate preferences towards certain nucleotides in the flanking sequence of the central C (Fig 2, "smoking" row; of note there is a slight preference towards upstream C). This is also reflected through the DNA shape, where the associations with mutational burden are likewise subtle when considered individually (Fig 3, "smoking" row). There are, for instance, positive associations of higher val-ues of propeller and roll parameters at positions 0 and +1, respectively, with mutation risk due to smoking chemicals. When considered jointly however, the DNA structural features do con-vey much information with predictive potential, which matches or exceeds the other consid-ered mutational signatures ($pR^2 = 0.85$ for the *5nt-str* approach, Table 1).

In addition to the chemical mutagenesis in lung, we also examined mutagenesis due to radiation (here, UV light) in a melanoma skin cancer genome using the Poisson regression analysis (Fig 2, "UV" row). It is known that pyrimidine base pair steps define the hotspots for electron excitation by UV light, leading to the formation of cyclobutane dimers [52–54]. The condition of having TC or CC steps for the excitation process is reflected in the known mutational signatures SBS7a and SBS7b, as well as in our regression analysis using trinucle-otides (*3nt-seq*, Fig 2A). Moreover, we observe the association of mutation risk with flank-ing T at positions -2 and +2 (*5nt-seq*, Fig 2B). This is consistent with previously reported pentanucleotide contexts for the C>T UV-associated mutations [55]. The DNA shape

analysis also reflects related trends: for example, a strong signal is observed for "*tw*-1*(H)*" (high twist at the -1 position), which is known to reach its highest values at TC base pair steps [32, 56]. Overall, the UV radiation induced mutation propensity of a site was highly predicable from the heptanucleotide-derived DNA shape features (*7nt-str*, $pR^2$ = 0.74, Table 1), similarly as for the DNA repair-related POLE (0.81) and APOBEC (0.85) mutagenesis. Thus, DNA shape features are useful to predict occurrence of mutations resulting from various causes, although we recognize this will not necessarily be the case for every mutagen to the same extent.

We examined a further chemical mutagenic agent by studying a temozolomide (TMZ)-treated glioblastoma tumor genome (Figs 2 and 3, TMZ row). TMZ is a DNA alkylating agent and its signature SBS11 has been detected in copious amounts in TMZ-pretreated tumors [1]. We note subtle associations with individual descriptors in all four groups of DNA shape features (*mgw*, *prop*, *roll* and *twist*; Fig 3). Compared to the other mutagenic agents considered, the DNA shape features considered jointly appeared less predictive for TMZ mutations (Table 1). This suggests that either DNA shape is less important for activity of this chemical mutagen, or that our set of shape features does not incorporate those features that are relevant for TMZ mutagenesis. However, an overall $pR^2$ of 0.52 (at *5nt-str*) implies there is still some signal relevant for predicting TMZ mutations embedded in the DNA shape descriptors.

A comparison of performance of the Poisson regression models in predicting mutability of oligonucleotides suggests that the *7nt-str* model tends to be more predictive (higher $pR^2$) than *5nt-str*, at least for four of the six considered types of hypermutation. This indicates that the shape of DNA in broader, heptanucleotide neighborhoods is relevant for the intensity of many mutational processes. The pentanucleotide-based DNA structure model has overall similar accuracy to the sequence-composition based (*3nt-seq* and *5nt-seq*) predictors, although their relative ranking is variable across different mutagens. Consistently across samples, we noticed that performance of the trinucleoide DNA sequence models (*3nt-seq*) outperforms performance of pentanucleotide sequence models (*5nt-seq*) (Tables 1 and S1). This may be due to data that are sparser, i.e. many more of the mutation count values in the regression tables for *5nt-seq* are zero, due to pentanucleotide motifs being longer and an individual pentanucleotide occurring less frequently genome-wide compared to an individual trinucleotide.

For each of the six types of hypermutators studied, we considered additional tumor genomes affected by the same mutational process. The outcomes of the analyses were consistent across individuals, in terms of the overall fit of regression models, both the structure-based and the sequence-based ones (S1 Table), with some variability noted across MSI samples. Furthermore, the statistical associations of individual structural features with mutation rates appeared overall consistent across tumors affected by the same hypermutation mechanism (S1 Fig). As a control, we considered tumor samples matched by tissue but not affected by the same hypermutation mechanism; instead, the mutations in these control samples likely originated from a mixture of similarly prevalent mutagenic mechanisms. As expected, the fit of the regression models–either sequence or structure-based one–was poorer on the control tumors (S1 Table). Additionally, these control samples did not exhibit the same patterns associations of structural features with mutation rates as the original hypermutator samples (S1 Fig).

Overall, the set of DNA shape features we examined is broadly reflective of oligonucleotide properties relevant for mutagenesis resulting from particular mutational processes. Nonetheless we recognize that this set of structural features may be further refined, thus possibly improving its accuracy in predicting DNA mutational hotspots for a broader range of mutagens.

## DNA shape features capture the variability in mutational exposures across individuals

Above, we have shown that DNA structure-based descriptors were associated with mutation risk resulting from exposure to diverse mutagens. Some of the shape features appeared to commonly predict higher mutation rates resulting from different mutagens (e.g., under-twisting at +1 position, Fig 3). However, many other features appeared associated with some mutagens but not with others (e.g., high roll at +1, low minor groove width at +1, high propeller at -1, etc.; see Fig 3). Thus, we hypothesized that DNA shape descriptions of mutated loci in a genome sequence could be used to quantitate mutagenic exposures that cell has undergone previously. In other words, use of DNA shape features would enable a novel conceptualization of mutational signatures [1, 2]. Mutational signatures are usually defined via trinucleotide neighborhoods of mutated loci, but broader neighborhoods were also considered previously (which however means that statistical interactions between nucleotides may need to be ignored) [57]. To derive mutational signatures, various forms of factor analysis can be applied to the mutation frequency data; commonly, non-negative matrix factorization (NMF) was used [2] although not exclusively [22]. Here, we evaluated the potential of DNA shape descriptors to generate informative mutational signatures. To this end we employed a principal components (PC) analysis to generate mutational signature PCs and measure the amount of systematic variability in the data, here implying the differential exposures to mutagenic effects across tumor samples. In the PC analysis, we used either the mutability of various DNA trinucleotides (sequence descriptors, *3nt-seq*) or mutability of various DNA shapes (*5nt-str* features; for the PC analyses, DNA shape features were adjusted to match the number of the DNA sequence features, see Methods for details). Both sets of features were included simultaneously in a PC analysis of somatic mutations from 1637 whole-genome sequences of tumors of various cancer types. Results showed that DNA structure features could explain a higher amount of the systematic variance than the DNA sequence features, when considering four out of the five dominant PCs: PC1, PC2, PC3, and PC5 (the PC analysis scree plot, broken down into the DNA sequence-feature part and the DNA shape-feature part, is shown in Fig 4A). Overall, the top five PCs explained 85.6% variance in the mutation rates of DNA oligonucleotides across tumors, with the sequence features covering 40.6%, and structural features 45.0% variance (Fig 4B). The excess of variance-explained by the structural features amongst the top PCs (Fig 4B) suggests they are more descriptive markers of the variability in mutagenic processes between human tumors, when compared to a standard DNA sequence representation using trinucleotides. Thus, DNA structural features are well-suited for inferring mutational signatures.

Over 45% of the variation that is explained by the dominant, first PC contains contributions from both the DNA shape features (23.7% variance) and from the DNA sequence features (21.8% variance). All mutation features had positive loadings on the PC1 (S1 Table), indicating that the PC1 reflects the overall mutation burden across tumors, rather than the differential mutational signatures. We noted a consistently high PC1 loading on the various types of mutations at loci with low DNA roll at -1 position ("ro1L"; S2 Table) suggesting this structural feature may be associated with higher DNA mutability generally i.e., in a manner not related to a particular mutagen. Furthermore, high-*mgw* and low-*twist* loci mutation frequencies had high PC1 loadings across multiple mutation types (S2 Table), suggesting additional structural features that characterize mutation-prone DNA loci, either due to being prone to various types of DNA damage, and/or less accurately copied or less efficiently repaired.

In PC2 and the following PCs (Fig 4C–4F), both positive and negative loadings of different DNA sequence features and structural features were observed. This means that these PCs can distinguish relative contributions of mutational processes across individual tumors i.e., the

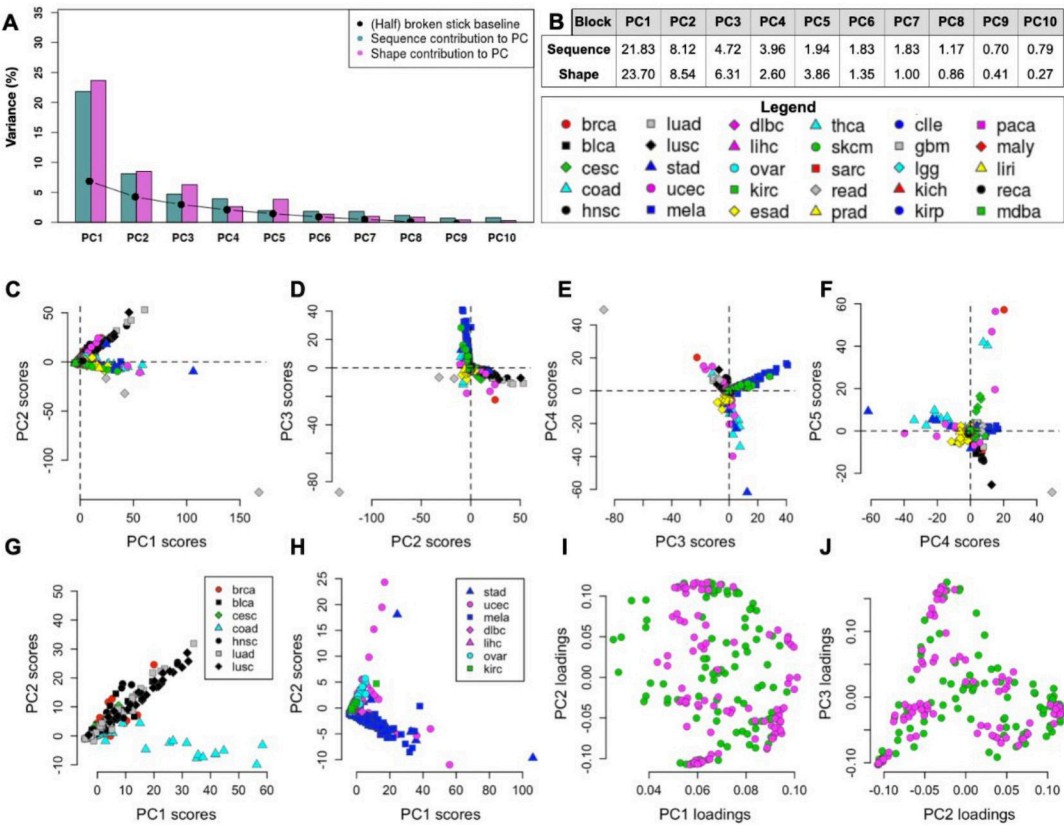

**Fig 4. PC analysis of mutation-associated DNA sequence and structural features in 1,637 cancer genomes in 30 cancer types.** (A) Top 10 PCs separated into sequence and structural contribution (baseline is the half of a 'broken stick' estimate). (B) Numerical representation of each PC's contribution to the overall variance in the sequence or structure block. (C) PC1 vs. PC2 scores, (D) PC2 vs. PC3 scores, (E) PC3 vs. PC4 scores, (F) PC4 vs. PC5 scores, (G-H) Examples of PC1 and PC2 scores of cancer samples separated into smaller groups of tissues, (I-J) Examples of loadings between PC1 –PC2 and PC2 –PC3 corresponding to the (C) and (D) PC scores.

PCs constitute mutational signatures in a broad sense. Expectedly, these PCs distinguish between abundance of six different mutation types across individuals, for instance PC2 contrasts genomes rich in C>G and/or C>A changes, from the genomes rich in A>C or A>G changes (see S2 Table). In addition to contrasting mutation types, the PC mutational signatures further distinguish between DNA structure and/or sequence contexts within each mutation type (see S2 Table). For instance, the PC2 –the dominant direction of differential mutability in our analysis–has high loadings (absolute value) for DNA under-twisting at the +1 position ("tw+1L" feature; S2 Table), mostly across the C>G and A>C mutation type. Thus, diverse types of structural features appear to describe the differential mutation rates of DNA oligomers across individuals. We note that DNA sequence features may also have high contributions to some of the top PCs (particularly PC4, PC7; Fig 4A), suggesting that both DNA sequence features and DNA structural features should be considered jointly when inferring mutational signatures.

To further examine how the dominant mutational processes (here: first two PCs) vary across different types of cancer, we divided the full dataset (from Fig 4C) into smaller groups containing diverse cancer types (Fig 4G and 4H; The PC loadings of each original variable are plotted in Fig 4I and 4J). Again, the PC1 components have only positive loadings, describing overall mutation burden, while the PC2 and following describe differential mutation rates

across DNA structural features (Fig 4I and 4J). In the various groupings of cancer types tested, the PC2 separates between cancer types: colorectal cancer *versus* others or uterus cancer *versus* others (Fig 4G and 4H), suggesting the ability of the DNA shape descriptors to capture tissue-specific mutational processes.

## Deriving mutational signatures by combining DNA trinucleotide sequence and DNA pentanucleotide structural features

The above analyses based on PC analysis suggest that DNA structure features of mutated loci can disentangle mutational processes affecting genomes of individuals. Mutational signature studies commonly employed NMF, a technique assuming additivity of the biological processes generating mutations [58]; this is seen as a desirable property for generating robust catalogs of mutagenic mechanisms. We adapted the NMF-based methodologies for mutation signature extraction (see Methods) to jointly incorporate a selected set of 96 oligonucleotide structural features, alongside the more standard set of 96 trinucleotide-based mutation types (16 trinucle-otides x 6 mutation types). Our method can extract signatures that resemble the known "SBS" (single-base substitution) signatures from the Catalogue of Somatic Mutations in Cancer (Cosmic) [59] in their trinucleotide spectrum, while additionally having a contribution from a set of DNA structural features, which may aid signature inference and interpretation. From the somatic mutations in 1637 considered cancer WGS, we recovered 48 such known signatures (matching one or more Cosmic SBS at a cosine similarity >0.85 in the trinucleotide spectrum). We further extracted 6 additional SBS-SS (structural signatures) which were novel i.e., their trinucleotide spectrum did not match a known Cosmic SBS, meaning that they were not previ-ously identified using trinucleotide analysis. The spectra of the composite structure-trinucleo-tide signatures are shown in Figs 5 and S3, and their exposures across cancer types in S4 Fig (underlying data for spectra and exposures is in S3 Table). While many extracted mutational signatures closely resembled their Cosmic counterparts in the trinucleotide spectrum, we rec-ognize that inference of some of known signatures may be able to be further improved. For example, the signature SBS29 was previously associated with tobacco chewing. In our analysis, SBS29L may be a mix of the two similar, C>A-rich signatures: the tobacco chewing SBS29, and the tobacco smoking signature SBS4 (judging by its exposure in two lung cancer types, S4 Fig). A further methodological consideration is that these NMF signatures were derived from raw mutation counts, thus their shapes in part reflect also the number of genomic nucleotides-at-risk encompassed by each feature in the spectrum (S5 Fig and S4 Table). This holds true both for the trinucleotide part of the spectrum (e.g. the NCG trinucleotides are rare in the human genome), and also for the structural features (e.g. *mgwL* bin, using the current defini-tion of *L*, *M* and *H* threshold, is rarer than *mgwM* and *mgwH* bins, S5 Fig). This may result in an apparent similarity of the profiles across different signatures.

Firstly, we examined the DNA structure part of the spectra of some of the known SBS signa-tures (S3 Fig), suggesting possible insights into mechanisms of mutagenesis. Several examples are highlighted below:

**SBS4** (tobacco smoking associated, likely resulting from bulky adducts e.g., by benzo[a]pyr-ene metabolites and related chemicals, typically onto guanine bases). This mutational process impacts regularly organized DNA structure, i.e., upon mutation, no specific deformations at C:G nucleotide pairs seem necessary for the mutagens to bind DNA. The less common muta-tions that occur at the A:T nucleotide pairs suggest slight preferences toward narrower minor groove and lower twisting at -1 position versus lower twist at +1 position, and higher propeller at the central base pair for A mutations to T or G respectively, suggesting an increased expo-sure at these sites.

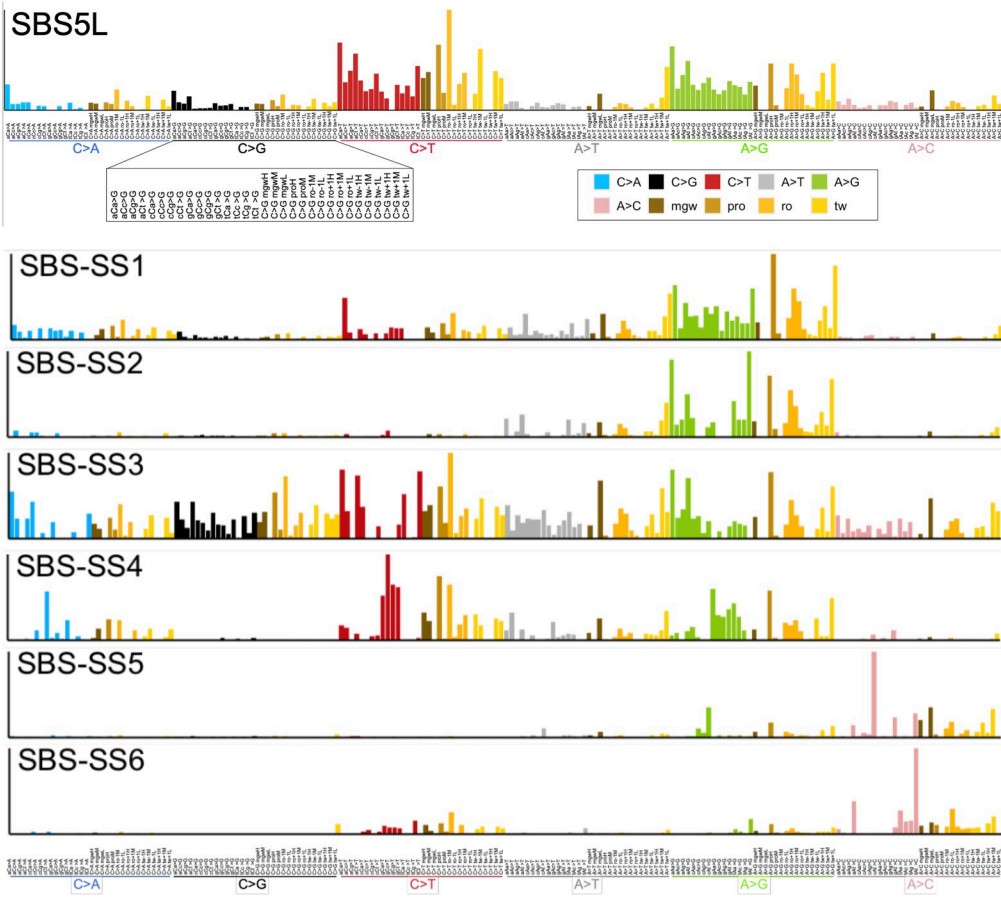

**Fig 5. NMF mutational signatures based on DNA sequence and structural features.** The composite mutational signatures consist of 96 DNA structural features: mgw and propeller at the central nucleotide (dark brown and light brown bars, respectively), roll and twist at -1 and +1 sites (dark yellow and light yellow bars, respectively), and the standard 96-component trinucleotide spectrum (blue, black, red, grey, green, pink bar colors). See S3 Fig for additional signatures extracted.

**SBS6/1L** (one of the signatures associated with DNA MMR failures). With the expanded minor groove, there is a potential for kinks occurring on DNA due to the positive or increased roll values at the adjacent bases, leading to a simultaneous compression of the major groove. It is conceivable that MMR proteins might act preferentially upon DNA with certain subtle conformational variations, increasing efficiency of repair at these loci.

**SBS 7a**, and similarly but to a lesser extent **SBS 7b** (UV exposure-associated mutational signatures, likely resulting from cyclobutane pyrimidine dimer formation). Series of positive roll values at the bases to the C>T mutation suggest a DNA structural motif with propensity toward the three-dimensional writhe, but a smooth curvature, which may either increase exposure to the damaging agent, or disfavor error-free lesion bypass or repair.

**SBS 10a** (associated with mutations in the proofreading domain of the replicative DNA polymerase epsilon). Signature 10 mutated loci have a characteristic combination of a high twist at -1 position followed by a low twist at +1 position, suggesting untwisting of DNA at the mutated base pair step immediately before overtwisting. This, together with a slight preference toward the *mgwH* and moderate roll values at the adjacent base pair steps, suggest a stronger kink or bending may occur. Such large conformational bias can conceivably impede replicative DNA polymerases, increasing mutation rates during their proofreading activity.

**SBS2** and **SBS13** (resulting from the activity of the APOBEC3A and/or APOBEC3B cytosine deaminases). The set of DNA shape descriptors employed results in a nondescript signal for these signatures, when compared with other mutators considered. Given that APOBECs are active exclusively on ssDNA, this relative lack of utility of double-stranded DNA shape information is perhaps expected to some extent and suggests that additional descriptors may be helpful to better model the APOBEC proteins recognition of DNA.

Next, we examined the spectra of 6 novel mutational structural signatures (SBS-SS) (Fig 5). These did not closely match an existing Cosmic SBS trinucleotide profile, suggesting that analysis of the DNA structural descriptors helps identify these novel signatures. We comment on these examples below:

**SBS-SS1** signature somewhat resembles signature SBS12 in the trinucleotide spectrum, in that they are enriched mainly in the A>G block, however SBS-SS1 does additionally encompass some C>A, C>T and A>T changes (Fig 5). The SS1 shape analysis suggests higher propensity for DNA bending caused by stretching of the minor grove (*mgwH*) supported by potential kinks due to high at roll +1 (*ro*+1H). Preference of low twist values at -1 and +1 additionally suggest unwinding of DNA, increasing the exposure of DNA bases. These structural changes are indication of deviation from the canonical B-DNA structure toward A-DNA. Such deviations could not be detected with the DNA trinucleotide signature analysis only. In terms of exposures, this SBS-SS1 is widespread across tissues with a somewhat similar distribution as SBS-SS3 (S4 Fig). This suggests that both SBS-SS1 and SBS-SS3 might have resulted from variations on a common mechanism of DNA replication or repair, present across various cell types.

**SBS-SS2** is characterized by mutations occurring mainly in the A>G block (Fig 5), where the trinucleotide spectrum (WAW, where W = A or T) resembles that of the error-prone DNA polymerase eta (POLH protein). This was seen in clustered mutational signatures in many cancer types, predominantly skin, liver, bladder, the digestive system and lung [51], as well as in the lymphoid cells because of the (non-pathogenic) process of somatic hypermutation. Similarly, the SBS-SS2 has the highest exposures in skin, liver, bladder and lymphocytes (S4 Fig), supporting the connection, and suggesting a structural basis for the mutability due to use of an error-prone DNA polymerase.

**SBS-SS3** has a recognizable structural feature across A>C and A>T blocks where the minor groove is compressed (*mgwL*, Fig 5). SS3 shows less DNA distortion in the A>G block, with a weaker signal for disrupted hydrogen bonding that can be recognized through the high values of propeller base pair parameter (*proH*). These observations suggest that within the SBS-SS3 signature, conformational DNA changes might occur on the larger scale (because of the compression of the minor groove repeated in two blocks, which might have an impact on the neighboring steps), without large displacements in the local base pair arrangements. With respect to the exposures (S4 Fig), SBS-SS3 is commonly observed across many different cancer types, with highest values in lung squamous cell carcinoma, in B-cell lymphoma, in ovarian, stomach and bladder cancers. The wide occurrence suggests this signature is due to a ubiquitous, endogenous process in replicating or repairing DNA, rather than an exposure to a particular carcinogen.

**SBS-SS4** signature's distinguishing feature is the proximity of low twist values in the adjacent base pairs associated with low roll values. Typically, the similar trends in twist and roll–either both low or both high–are rarely observed at the same base pair step, suggesting prevalence of purine-purine (RR) motifs. The presence of RR steps is supported by the signals observed in the SBS-SS4 trinucleotide spectra where a higher mutational frequency of GAA, GAC, GAG, GAT and TAA sequences is detected.

**SBS-SS5** displays a variety of structural geometries depending on which type of A>N mutation is considered; the A>C changes appear dominant (Fig 5). Within the A>C block there is a preference for either compressed or extended minor groove width, with a lower preference toward regular width. This changes when mutations occur in the A>T or A>G block: compressed or extended minor groove widths (*mgwL*, *mgwH*) are preferred, respectively. These observations suggest influence of both DNA sequence and the DNA conformation on the recognition of DNA by the mutagenic agents. The high exposure of this signature in the esophagus, stomach, colon and lymphoma (S4 Fig) suggests a relationship with the known signature SBS17b (which is also A>C rich and occurs in these tissues) however the trinucleotide spectra are sufficiently different that it was identified as a separate SBS.

**SBS-SS6** contains mutations of the A>C and C>T types (Fig 5) and is rarely occurring (S4 Fig). The strong compression of the central minor groove might suggest a preference for helix-turn-helix binding protein domains, or for proteins further compressing the minor groove, yet the preference for the simultaneous expansion of minor groove is intriguing and could be further explained by different content of purines and pyrimidines in the adjoining bases.

## Discussion

Our work highlights the ability of DNA shape features to predict mutational risk of individual genomic loci in cancers exposed to various mutagenic processes, ranging from DNA repair failures to mutagenic chemicals or radiation. Our predictive models further showcase the ability to identify DNA structural determinants associated with each mutagenic process. Many of the known trinucleotide mutational signatures appear to have informative structural components of the spectra. Such associations of mutation risk with DNA structural features may further our understanding of the underlying mechanisms of DNA damage and repair. Furthermore, we demonstrate that the DNA shape features of the mutated loci can capture a higher amount of systematic variability in mutational processes across cancer samples than a naive representation of DNA sequence via oligonucleotides, as commonly employed. Consistently, using structural features, novel mutational signatures can be extracted that may not be within reach of oligonucleotide-based approaches. We note that these two groups of features (DNA shape and DNA sequence) are to some level redundant–expectedly so, given that the shape features were derived using a method based on oligonucleotide dictionaries. Importantly, however, the DNA shape features represent a tradeoff between complexity and informativeness; they capture certain interactions across neighboring nucleotides within a pentanucleotide or longer neighborhood, while keeping the overall representation relatively simple (not all statistical interactions between nucleotides are modeled). This makes DNA structure useful for analysis of—relatively sparse—mutation count data. If the full set of statistical interactions was included for DNA sequence representations, the number of features would rise exponentially, making it unfeasible to apply to contexts longer than pentanucleotides (we note that for the exome sequencing data, which is most abundant, even the pentanucleotides are out of reach because of small number of mutations per tumor). Use of structural DNA features however makes such analyses of broader contexts feasible. This is clearly of interest, given that heptanucleotide and even nonanucleotide contexts appear relevant for some mutagenic processes, such as UV mutagenesis [55], POLE mutagenesis in cancers [15], and also certain mutational processes in the human germline [60, 61]. Overall, we suggest that use of structural DNA features may help overcome hurdles for analyzing the roles of longer oligonucleotide neighborhoods as determinants of DNA mutability.

One caveat of the set of structural features currently employed is that their intended use is for ds DNA, however they are ill-defined in ss DNA segments. This means that the framework

presented here, in its current implementation, would not be able to adequately address those mutational processes mediated by structured ss DNA. One known example are the hypermutable hairpin structures in DNA, where the loop is a target of the APOBEC3A cytosine deaminase [25] and is thus hypermutable (we note that only a minority of APOBEC3A mutations genome-wide is mediated by these stem-loop structures [62]). Future work building upon our study may incorporate DNA structural descriptors pertinent to ss DNA, as well as increase the DNA oligomer length by drawing upon the ever-increasing amounts of cancer genomic data to increase statistical power.

There is a growing awareness that analysis of somatic mutation data may be able to provide markers to guide personalized therapy for cancer patients. Current focus is on refining drug selection to match the underlying genetic profile of tumors, particularly in terms of driver mutations [63–65]. However, because mutational signatures reflect ongoing genomic instability, which is common in tumors and recognized as a targetable vulnerability of cancer, mutational signatures could help better stratify patients for targeted therapies e.g. immunotherapy for MMR-deficient tumors [66, 67]. The mutational Signature SBS3 and also a pattern of deletions with microhomology signal failures in the homologous recombination repair (HRR) pathway, which is targetable by PARP inhibitors [19, 20]. This principle may extend beyond MMR and HRR deficiencies: in an analysis of mutational signatures across cancer cell line panels screened for drug activity, many signatures were associated with drug response, suggesting that some of the signatures might constitute useful genomic markers in patients [21]. For example, given that the error-prone DNA polymerase eta (POLH protein) can cause resistance to treatment with cisplatin [68], identification of POLH mutational signatures–previously via DNA sequence features [51] but possibly also by the affinities of this error-prone DNA polymerase(s) towards certain DNA shape features (Fig 5)–could inform treatment decisions. In summary, there is great promise for clinical use of genomic predictive markers in tumors even though often the underlying mechanisms remain elusive [65]. Among such genomic markers, the utility of mutational processes in particular merits more attention.

With a joint DNA sequence and structure representation as a basis for mutation signature inference, we posit that future work will be able to disentangle the mutagenic mechanisms at increased accuracy compared to current, oligonucleotide-based representations. Moving beyond examining the relatively simple mutational landscapes of hypermutator tumor samples–as we have done here to establish a proof-of-principle–future methods to predict mutation risk of genomic loci should also be able to accurately deconvolute mutational processes in tumors where more than one process was active. Next, our current implementation of the multi-tumor analysis to extract mutational signatures is based on a broadly standard NMF approach. Conceivably, this may further benefit from applications of sophisticated statistical approaches for mutational signature deconvolution that use e.g. topic models [69, 70] and that can jointly analyze multiple 'channels' in the mutational signature signal. In conclusion, we propose a general framework for quantifying mutational signatures via use of DNA shape descriptors, which may advance mechanistic understanding of mutagenesis and identify novel processes shaping mutational landscapes across individuals.

## Supporting information

**S1 Fig. Poisson regression coefficients for 3nt-seq, 5nt-seq, and 5nt-str representations, for additional tumor samples.** For the sequence block, the four DNA bases A, C, G, or T are color coded. For the structural block, parameters are normalized and divided into 3 equally spaced bins: high (light pink), medium (medium), and low (dark). Error bars superimposed

on each symbol show 95% C.I.
(TIFF)

**S2 Fig. Poisson regression coefficients for APOBEC-tumor samples separated by C>G and C>T mutation types.** Shown for the (A) 3nt-seq and (B) 5nt-str models.
(TIFF)

**S3 Fig. NMF mutational signatures based on DNA sequence and structural features.** The composite mutational signatures consist of 96 DNA structural features: mgw and propeller at the central nucleotide (dark brown and light brown bars, respectively), roll and twist at -1 and +1 sites (dark yellow and light yellow bars, respectively), and the standard 96-component tri-nucleotide spectrum (blue, black, red, grey, green, pink bar colors).
(TIFF)

**S4 Fig. The exposures of structural mutational signatures from Figs 5 and S3.** The mean exposure of all tumor samples in that cancer type is shown.
(TIFF)

**S5 Fig.** Number of nucleotides-at-risk in the human genome that may be affected by mutations, stratified by trinucleotide context (A) or by structural features (B).
(TIFF)

**S1 Table. Fit of the Poisson regression models for additional examples of tumors affected by the mutagenic processes studied.**
(DOCX)

**S2 Table. PCA analysis combining sequence features and structural features of DNA oligomers.**
(XLSX)

**S3 Table. NMF analysis generating structural mutational signatures.** Sheets within the table contain signature spectra, exposures per tumor sample, and normalized exposures.
(XLSX)

**S4 Table. Normalized counts of nucleotides-at-risk, stratified by trinucleotide and by DNA oligomer structural features.**
(XLSX)

## Author Contributions

**Conceptualization:** Aleksandra Karolak, Fran Supek.

**Data curation:** Aleksandra Karolak, Fran Supek.

**Formal analysis:** Aleksandra Karolak, Jurica Levatić.

**Funding acquisition:** Fran Supek.

**Investigation:** Aleksandra Karolak.

**Methodology:** Aleksandra Karolak, Jurica Levatić, Fran Supek.

**Project administration:** Fran Supek.

**Software:** Aleksandra Karolak, Jurica Levatić.

**Supervision:** Fran Supek.

**Validation:** Aleksandra Karolak.

**Visualization:** Aleksandra Karolak, Jurica Levatić.

**Writing – original draft:** Aleksandra Karolak, Jurica Levatić, Fran Supek.

**Writing – review & editing:** Aleksandra Karolak, Fran Supek.

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
