## [Decision Letter · Decision Letter 0]

2 Nov 2021

PONE-D-21-30926A framework for mutational signature analysis based on DNA shape parametersPLOS ONE

Dear Dr. Supek,

Thank you for submitting your manuscript to PLOS ONE. After careful consideration, we feel that it has merit but does not fully meet PLOS ONE’s publication criteria as it currently stands. Therefore, we invite you to submit a revised version of the manuscript that addresses the points raised during the review process.

While both reviews are favorable, there are multiple concerns about several aspects of statistical analysis, approach, interpretation and presentation. Please, address all reviewer's comments in a revised submission.

We look forward to receiving your revised manuscript.

Kind regards,

Sergey Korolev, Ph.D.

Academic Editor

PLOS ONE

Journal Requirements:

"This work has received funding from the European Union’s Framework Programme for Research and Innovation Horizon 2020 (2014-2020), under the Marie Skłodowska-Curie PROBIST grant agreement No. 754510 (to A.K. and J.L., PROBIST co-fund fellowship of the Barcelona Institute of Science and Technology) and by the European Research Council (ERC) under the European Union’s Horizon 2020 research and innovation programme (grant agreement No. 757700, to F.S). F.S. is funded by the ICREA Research Professor program. A.K., J.L. and F.S. acknowledge support of the Severo Ochoa Centres of Excellence program of the Spanish Ministry of Economy and Competitiveness to the IRB Barcelona. Work in the laboratory of F.S. was supported by the ERDF/Spanish Ministry of Science, Innovation and Universities-Spanish Research State Agency/RegioMut project (grant agreement No. BFU2017-89833-P). "

"This work has received funding from the European Union’s Framework Programme for Research and Innovation Horizon 2020 (2014-2020), under the Marie Skłodowska-Curie PROBIST grant agreement No. 754510 (to A.K. and J.L., PROBIST co-fund fellowship of the Barcelona Institute of Science and Technology) and by the European Research Council (ERC) under the European Union’s Horizon 2020 research and innovation programme (grant agreement No. 757700, to F.S). F.S. is funded by the ICREA 

Research Professor program. A.K., J.L. and F.S. acknowledge support of the Severo Ochoa Centres of Excellence program of the Spanish Ministry of Economy and Competitiveness to the IRB Barcelona. Work in the laboratory of F.S. was supported by the ERDF/Spanish Ministry of Science, Innovation and Universities-Spanish Research State Agency/RegioMut project (grant agreement No. BFU2017-89833-P).

Reviewers' comments:

Reviewer's Responses to Questions

**Comments to the Author**

1. Is the manuscript technically sound, and do the data support the conclusions?

Reviewer #1: Yes

Reviewer #2: Yes

2. Has the statistical analysis been performed appropriately and rigorously? 

Reviewer #1: Yes

Reviewer #2: Yes

3. Have the authors made all data underlying the findings in their manuscript fully available?

Reviewer #1: Yes

Reviewer #2: Yes

4. Is the manuscript presented in an intelligible fashion and written in standard English?

Reviewer #1: Yes

Reviewer #2: Yes

5. Review Comments to the Author

Reviewer #1: Mutational signatures represent different mutational processes underlying cancer initiation and development, and they are usually reported by the frequency of tri/penta-nucleotide contexts. This study considers structural features that may enhance the interpretation of sequence-based signature. Overall, this work is of high quality. Although the ability of shape parameters to predict mutation probability is not better than sequence context, the incorporation of trinucleotide-based mutation type with structural-based parameters could provide new insights for mechanisms of mutagenesis.

I have below comments.

1, The author should explain why only focus on these three shape parameters. They mentioned the affect of hairpin structure on mutation formation, but not actually include it as a parameter.

2, For Fig2 and Fig3, the author only used one sample to represent the typical mutational processes. It would be good to see if the trend still holds for additional individuals as there are a lot of such samples available from TCGA/ICGC. In addition, POLE mutants have different hotspots (like V411L and P286R) with differential spectrum (Fang et al, 2020, PLos Gene; Hodel et al, 2020, Mol Cell), the sample used in this study should be indicated.

3, Poisson regression to evaluate sequence-based variables and structural-based variables for mutation probability would be good to be presented by an equation in the method.

4, In page 34, I think the second paragraph is out of scope of this paper. Maybe it is better to discuss how the shape parameters incorporated-signature would benefit to personalized therapy.

Reviewer #2: In this study Karolak et al. described how data on DNA structure can predict mutation probability. They firstly analyzed the role of DNA structure and nucleotide context in mutation probability independently and then tried to extend the idea of standard mutational signatures (based on trinucleotide mutational spectrum) by including information about structural features. The idea itself looks very exciting. My main concern about it is prediction of structural features from DNA sequence and very strong correspondence between contexts and structures at such a small scale. Mutagenic processes in investigated cancers are highly context-dependent, so by definition you will have structure-dependence as well if each context corresponds to very specific structure. It would be informative to understand how much different contexts vary in described structural parameters. It would also be interesting to see whether different contexts with similar structure can have similar changes in mutation probability. Probably it would be helpful to investigate more large-scale structures to see whether there are differences in mutability of particular context in different structures? Similar to example with APOBEC in stem loops.

For investigated cancers, models with structural features performed similarly well to contexts. It would be interesting to investigate whether this is the case for cancers without such high context-specificity.

The most interesting part of the paper is prediction of new signatures that could not be extracted based only on nucleotide context. Although the method seems not to be ideal and the biological interpretation is quite limited, this work is a good start point in this direction.

Comments:

1. Figure 1 is not fully understandable. From the scheme, it seems that far right and far left nucleotides are not engaged in any parameter estimation that seems not to be true.

2. Is it correct that Poisson regressions do not take information about mutation type into account? I mean that coefficients on Figure 2 and Figure 3 show influence on mutation probability of central nucleotide independently of mutation type. In this case, they seem to be biased by the main mutation type in cancer.

3. Could you please discuss in text why 3nt-seq model always performs better than 5nt-seq model? (Table 1).

4. Page 13 “the structural DNA features compared favorably (pR2=0.81 for 7nt-str, Table 1) to the sequence DNA features (pR2=0.81 for 5nt-seq)”. Why do you call this comparison favorable if they have equal pR2?

5. Page 13 “The overall predictive accuracy was higher for the POLE tumor than for MSI tumor (Table 1), suggesting that mutational risk can be predicted from DNA shape to a variable degree across different mutational processes.” From Table 1 it is seen that context-model also performs better for PolE compared to MSI. As structural features were predicted from nucleotide sequence this fact can probably be explained by lower context-specificity of MMR compared to PolE.

6. Page 14 “Considered together with over- and under-twisting at positions -1 and +1, respectively, this suggests that DNA motifs experiencing APOBEC mutagenesis may be more prone to have flipped out bases”. It is interesting to investigate the co-occurrence and simultaneous influence of structures found significant in Poisson regression.

7. Figure 5. X-axis is impossible to read. Structural signatures seems to be very similar between different signatures in contrast to difference between context signatures. For example, structural elements for SBS1, SBS3L, SBS6, SBS7b seems very similar for C>T mutation but contexts differ extremely. Could you explain this? I lack the interpretation as I was not able to understand the correspondence of bars to structural features.

8. Figure 6 looks confusing. For example, SBS4 that you describe as tobacco smoking induced was not found in any lung cancer. For LUSC you determined high exposure to SBS8L/4L that is also similar to SBS4 but for luad you didn’t observe any of them. SBS7a,b,c are not pronounced in skcm.

Misprint

1. Page 19 “suggesting that both DNA sequence features and the and DNA structural features should be considered jointly…”

6. PLOS authors have the option to publish the peer review history of their article (what does this mean?). If published, this will include your full peer review and any attached files.

Reviewer #1: No

Reviewer #2: No

---

## [Author Response · Author response to Decision Letter 0]

16 Dec 2021

We have attached the response-to-reviewers PDF, containing a point-by-point response to queries and an overview of changes to the manuscript.

---

## [Decision Letter · Decision Letter 1]

27 Dec 2021

A framework for mutational signature analysis based on DNA shape parameters

PONE-D-21-30926R1

Dear Dr. Supek,

We’re pleased to inform you that your manuscript has been judged scientifically suitable for publication and will be formally accepted for publication once it meets all outstanding technical requirements.

Kind regards,

Sergey Korolev, Ph.D.

Academic Editor

PLOS ONE

Additional Editor Comments (optional):

Reviewers' comments:

Reviewer's Responses to Questions

**Comments to the Author**

1. If the authors have adequately addressed your comments raised in a previous round of review and you feel that this manuscript is now acceptable for publication, you may indicate that here to bypass the “Comments to the Author” section, enter your conflict of interest statement in the “Confidential to Editor” section, and submit your "Accept" recommendation.

Reviewer #1: All comments have been addressed

Reviewer #2: All comments have been addressed

2. Is the manuscript technically sound, and do the data support the conclusions?

Reviewer #1: Yes

Reviewer #2: (No Response)

3. Has the statistical analysis been performed appropriately and rigorously? 

Reviewer #1: Yes

Reviewer #2: (No Response)

4. Have the authors made all data underlying the findings in their manuscript fully available?

Reviewer #1: Yes

Reviewer #2: (No Response)

5. Is the manuscript presented in an intelligible fashion and written in standard English?

Reviewer #1: Yes

Reviewer #2: (No Response)

6. Review Comments to the Author

Reviewer #1: (No Response)

Reviewer #2: (No Response)

7. PLOS authors have the option to publish the peer review history of their article (what does this mean?). If published, this will include your full peer review and any attached files.

Reviewer #1: No

Reviewer #2: No

---

## [Editor Report · Acceptance letter]

3 Jan 2022

PONE-D-21-30926R1 

A framework for mutational signature analysis based on DNA shape parameters 

Dear Dr. Supek:

I'm pleased to inform you that your manuscript has been deemed suitable for publication in PLOS ONE. Congratulations! Your manuscript is now with our production department. 

Kind regards, 

on behalf of

Dr. Sergey Korolev 

Academic Editor

PLOS ONE